# Addressing Ancestral Underrepresentation in Oncobiology: The Need for Sub-Saharan African-Specific In Vitro Models

**DOI:** 10.3390/genes16121403

**Published:** 2025-11-24

**Authors:** Carla S. dos Santos, Ana C. Magalhães, Ricardo J. Pinto, Carla Carrilho, Cláudia Pereira, Fernando Miguel, Pamela Borges, Lúcio Lara Santos, Luisa Pereira

**Affiliations:** 1i3S—Instituto de Investigação e Inovação em Saúde, Universidade do Porto, 4200-135 Porto, Portugal; 2IPATIMUP—Instituto de Patologia e Imunologia Molecular da Universidade do Porto, 4200-135 Porto, Portugal; 3ICBAS—Instituto de Ciências Biomédicas Abel Salazar, Universidade do Porto, 4050-313 Porto, Portugal; 4Department of Pathology, Faculty of Medicine, Eduardo Mondlane University, Maputo 2HJP+QWV, Mozambique; 5Department o Pathology, Maputo Central Hospital, Maputo 2HJR+JFQ, Mozambique; 6Unidade Local de Saúde Almada-Seixal, 2805-267 Almada, Portugal; 7Faculdade de Medicina, Universidade de Lisboa, 1649-028 Lisboa, Portugal; 8Angolan Institute Against Cancer, Luanda 56FG+7R6, Angola; 9Molecular Biology Laboratory, Hospital Universitário Agostinho Neto, Praia 112, Cape Verde; 10Grupo de Patologia e Terapêutica Experimental e Departamento de Oncologia do Instituto Português de Oncologia do Porto, 4200-072 Porto, Portugal; 11ONCOCIR—Education and Care in Oncology, PALOP—Lusophone Africa, 4200-072 Porto, Portugal

**Keywords:** Sub-Saharan Africa, oncology research, in vitro models, cancer cell lines, conditional reprogramming

## Abstract

Cancer is an increasing public health burden, including in Sub-Saharan African (SSA) populations, where cancer incidence is predicted to increase by around 140% between 2022 and 2050. These rates require a better understanding of the epidemiological, clinical, and genetic/molecular characteristics of cancer in SSA populations. There is an urgent need to improve the genomic characterization of SSA tumour samples and also to establish suitable in vitro models for hypothesis testing. In fact, even though thousands of cancer cell lines (CCLs) have been established employing different methods of cell immortalization and have been included in deep molecular characterization panels, SSA ancestry is limited to only ~6% (mostly African Americans, who represent limited diversity in the context of the African continent) of publicly available CCLs. This disparity needs to be addressed by using next-generation immortalization methods such as conditional reprogramming to establish CCLs derived from SSA cancer patients that also represent the diversity within the African continent. Research in SSA oncobiology has the potential to add essential information to better understand the diverse molecular pathways leading to cancer and to find promising therapeutic avenues. We also discuss the challenges to conducting oncobiology studies with cell modelling derived from SSA patients in low-to-middle-income African countries, such as Portuguese-speaking African countries.

## 1. Introduction

Sub-Saharan African (SSA) populations have had the deepest evolutionary history of our species since the emergence of *Homo sapiens* on this continent around 300,000 years ago (reviewed in [1]). This fact led to high genetic diversity and considerable population structure across the African continent. Also of importance is the fact that positively selected variants have increased in frequency due to environmental adaptations, impacting health liabilities [1,2,3]. In past and current times, exposures to novel pathogens and changes in lifestyle have been paramount in human adaptation [4]. Although infectious diseases still dominate African health agendas, non-communicable diseases are on the rise, including cancer [5]. It is predicted [6] that, if no intervention is made at the moment, SSA will experience a major increase in cancer mortality from 763,843 annual deaths in 2022 to almost 1.8 million by 2050.

In general, SSA populations are understudied in terms of genomics when compared with their counterparts of European and Asian ancestry, which limits the transferability of findings to the African context. This is true for global population diversity [1,7] and for disease cohorts [8], including cancer [9,10]. Consequently, the discoverability potential from SSA genomic studies is unique at the global scale and will provide powerful insights allowing us to understand the genomic and molecular basis of health and disease [11]. An illustrative example in cancer is how the high prevalence of Burkitt’s lymphoma in Africans helped to identify Epstein–Barr virus as its causal factor (reviewed in [12]). Given the high incidence of infection-related [13] or infection-exacerbated cancers in SSA [14], the potential for findings related to immune-related issues in oncobiology is quite promising.

While researchers have been calling for action in the characterization of genomic data for SSA and have begun to address this gap [15,16,17,18], progress is still lagging in the establishment of representative cell models for African continental genomic diversity. Genomic findings must be evaluated through functional assays conducted in relevant cell models to probe the dialogue between genotypes and phenotypes [19]. Without representative cell models for SSA, the discoveries that are beginning to emerge from omics studies will remain unaddressable functionally or at most addressed with limited resolution [20]. In this review, we reflect on this serious need for representative cell models for SSA and on the technological advances in their establishment in recent years. These improvements should be put into practice to establish up-to-date in vitro models that enable oncobiology research and translation for individuals with SSA ancestry.

## 2. Cancer in SSA

Most estimates of the cancer increase in SSA are dire. The incidence of cancer has doubled in the past 30 years, and this rate can double again in only 20 years if no action is taken at the moment (reviewed in [5]). The 1,884,322 annual new cancer cases in SSA in 2022 will rise above 2.8 million in 2050 [6]. This dramatic increase in SSA cancer incidence is mainly due to a social and economic shift [21]. Health care is improving in Africa, leading to an increase in life expectancy, with the number of cancer cases being a consequence of this [22]. Additionally, changes in population habits like smoking, alcohol consumption, diet based on processed food and meat, and overall urbanization are leading to a rise in the exposure to carcinogens and mutagens and consequently a higher cancer risk [21,22].

Data from Globocan 2022 [6] show differences in cancer incidence and mortality between SSA and high-income countries (HICs; Figure 1). In SSA, the incidence of prostate and cervical cancer is predominant, in opposition to the predominance of lung and colorectal cancers in HICs. What is particularly interesting is that infectious agents are a significant cause of cancer in SSA. Parkin et al. [13] estimated that 28.7% of cancers in SSA are due to infectious agents, especially viruses, such as human papillomavirus (HPV), herpesvirus, hepatitis viruses (B and C), Epstein–Barr virus, and human immunodeficiency virus, but also *Helicobacter pylori* and *Schistosoma haematobium*. This infection-related proportion is twice as high as the value of 12.5% of cancers associated with traditional risk factors such as smoking, alcohol, and unhealthy diet [13], although this last fraction is probably on the rise in Africa given the ongoing lifestyle changes.

Not only are there differences in cancer types between SSA and HICs, but there are also differences in the subtypes of specific cancers. The best example is triple-negative breast cancer (TNBC), which has a high prevalence in SSA women [23,24], with values as high as 40%, as opposed to 10–20% in European women [25]. It also affects women of African descent on other continents at a younger age: in the USA, the mortality rate due to BC in AA women younger than 50 years old is 77% higher when compared with European American (EA [25]) patients. This higher mortality rate is related to the cancer’s more invasive nature, the high probability of metastasis, and the absence of a targeted treatment for TNBC [26].

Also interesting is the fact that the incidence rates of cancer types between SSA regions can be significantly diverse. A paradigmatic example is the so-called African oesophageal cancer corridor, largely overlapping with the Great Rift Valley (broadly East Africa), where the squamous subtype reaches very high incidence rates (reviewed in [27]). Several risk factors, from environmental (selenium and zinc deficiencies, presence of mycotoxins) to behavioural (tobacco and alcohol consumption, hot foods and beverages, use of wood for cooking) ones, as well as genetic and microbiome diversities, can possibly explain this differential burden within the African continent, but a systematic evaluation has not yet been conducted. Another example is the extremely high rate of infection due to *H. pylori* (~95%), combined with very low levels of gastric cancer [28], in the Mozambican population in comparison with other SSA countries such as Angola, where this cancer is more frequent [22,29]. The high infection rate and low gastric cancer rate observed in Mozambique suggests a co-evolution between this bacterium and SSA human ancestry, a hypothesis that is beginning to be supported by epidemiological [30] and incipient in vitro testing [20]. The higher European admixture in the population of the Angolan capital in relation to the population of the Mozambican capital (our own unpublished data) can contribute to the observed differences in gastric cancer incidence, along with other risk factors.

Inadequate prevention and delayed diagnosis lead to most cancers being diagnosed at a late stage, which, coupled with inadequate treatment options, causes cancer mortality rates to be 1.5- to 4-fold higher in Africa than in HICs [21]. More recent calculations [6] indicate a 0.66 mortality-to-incidence ratio in SSA compared with 0.10 in HICs, resulting in a 6.6-fold higher cancer mortality rate. The mortality rates in SSA are also high in cancer types that have a good survival rate in the rest of the globe, such as prostate, breast, and colorectal cancer. Promisingly, a huge impact on the SSA cancer burden could be easily and rapidly obtained through widespread vaccination against HPV and hepatitis B and through treatment with antibiotics for *H. pylori* and praziquantel for schistosomiasis [5]. A successful current example is the near eradication of HPV, and hence of cervical cancer, in Rwanda due to an efficient implementation of a national HPV vaccination programme [31]. Hopefully, as other SSA countries begin improving their vaccination programmes [5], a significant impact will be observable in the next years in the SSA cancer registry conducted by the African Cancer Registry Network (AFCRN—https://afcrn.org/index.php; accessed on 17 November 2025).

## 3. Cancer Genomics in People of SSA Ancestry

In relation to the discovery of genetic factors increasing the risk of developing cancer, genome-wide association studies (GWASs) have been making a difference [32]. These studies resort to high-throughput genetic technology that allows to genotype thousands or millions of variants at once, usually through the use of microarray technology [33]. Park and co-authors [10] compiled cancer GWASs from the literature and repositories published till the end of 2016 and concluded that more than 700 loci involved in cancer risk were identified, but the ancestry of the samples included was largely European: 84% European, 11% East Asian, 4% African, and 1% Latin American. The poor representation of African ancestry in cancer GWASs follows the global GWAS pattern [8], with only 2.4% of screened individuals being of this ancestry. Despite the low inclusion of SSA individuals in GWASs, the SSA contribution to disease-related genetic associations is surprisingly high, at 7%. This great potential for discovery in SSA studies reflects the fact that an African genome has on average 20% more variants compared to European and East Asian genomes [34]. Hence, a significant proportion of genetic diversity present in SSA populations continues not to be probed in research studies. It is not surprising that this under-representativeness of non-European populations renders European-based findings non-translatable to other ethnicities [35]. In the context of cancer, this fact will limit the transferability of preventive cancer strategies and personalized cancer therapies.

Specific SSA genetic cancer signatures are observed when characterizing tumour samples. The Cancer Genome Atlas Program (TCGA; [36]) performed the molecular characterization of over 20,000 primary cancers and matched normal samples from 33 cancer types. As many of the samples were from patients from the USA, omics are available for AA patients, mostly limited to cancer types highly prevalent in this ancestry, such as BC and prostate cancer [37]. For prostate cancer, Mori et al. [38] identified genes of the keratin family with higher expression in AAs, also linked to oncogenic gene signatures (enriched KRAS and ERBB2 pathways) and to basal and luminal A subtypes (with poor postoperative androgen deprivation therapy response). Krishnan et al. [39] found that in TCGA of clear cell renal carcinoma, the AA cohort has a lower mutation rate of the tumour suppressor gene *VHL* and a correspondingly lower level of HIF-1α/VEGF pathway activation, explaining the resistance of these patients to the commonly used VEGF-targeted therapies.

AA origin dominates the scarce SSA genomic tumour data, but AA diversity is a limited proxy of the overall continental African genetic diversity [16] and is biased in terms of environmental and epidemiological cancer risks. Genomic characterizations of SSA cancer cases are finally beginning to be published [40,41,42,43] and are offering interesting insights. A multi-omics study in Nigerian BC cases [41] led to the identification of a specific mutational profile that was distinct from the AA profile and compatible with a more aggressive biological behaviour: increased mutational signature associated with deficiency of the homologous recombination DNA repair pathway; pervasive mutations in the *TP53* gene; mutations in the *GATA3* gene; and greater genomic mutational burden. Also, a careful catalogue of *BRCA1* and *BRCA2* mutations in multi-ethnic families [44] showed specific founder mutations in AA, Nigerian, and South African families. An exomic plus untranslated region characterization of Angolan and Cape Verdean TNBC cases [40] confirmed a high somatic burden, with 86% of variants being unreported so far, with some predicted deleterious variants located on known cancer driver genes (especially *TP53*) but indicating possible novel TNBC driver genes that may play a major role in the African context, such as *TTN*, *CEACAM7*, *DEFB132*, *COPZ2*, and *GAS1*. Thus, the genetic characterization of SSA cancer samples is of the utmost importance.

## 4. In Vitro Models

Given the specific epidemiologic and genetic cancer signatures that are emerging for the SSA ancestry, representative models of this ancestry must be established to test the hypotheses. In vitro testing, involving components of an organism, such as cells or molecules and, more recently, organoids, is the first step in the functional assessment of genomic evidence. So far, most in vitro functional assays in the cancer context are being addressed without taking into account the potential effect of ancestry. As shown before in this review, there is evidence that ancestry adds a new level of complexity to the equation of cancer heterogeneity and, as such, it is an important co-variable to be taken into consideration. But the field still lacks cell models representative of human ethnicities/main population groups.

### 4.1. Cancer Cell Lines (CCLs)

Basic cancer testing and understanding have been accomplished, over the past 70 years, by recurring to CCLs (reviewed in [45]). CCL testing has also been essential in the context of other diseases, to test infection due to pathogens, and for drug screenings [45,46]. Since the establishment of the first human CCL in 1951, thousands of CCLs have become available, mainly in the 1970s/80s/90s, and have been included in repositories [47] and panels [48]. More recently, these CCL panels have been characterized in detail for multi-omics information [45].

The processes needed to establish a CCL are not trivial, and several problems can occur when establishing and using cell lines, such as cell line misidentification, contamination with microorganisms (e.g., mycoplasma), and genetic and phenotypic instability, which are minimized by following sets of good practices [47]. The difficulty of establishing a CCL is somewhat puzzling since cancer cells evade cell cycle and death control mechanisms and are characterized by fast proliferation. However, most times, cancer cells cannot proliferate and die after a few weeks in an in vitro environment. Even when a CCL is established, the process to certify real immortalization is laborious [47,49,50]. The United Kingdom Coordinating Committee on Cancer Research (UKCCCR) guidelines [47] suggest a detailed characterization, confirmation of the immortalization of the culture, proof of neoplasticity, authentication of the true origin of the cells, scientific significance, and availability of the cell line for other investigators, among other warnings and recommendations.

Despite these difficulties, the impact of CCLs in the advancement of knowledge on oncobiology is highly recognized. And the field is moving towards advanced in vitro models that may better mimic living human conditions.

#### 4.1.1. Historical Contextualization

The first human CCL, known as HeLa, was established in 1951, with cells from a cervical cancer biopsy isolated by George Gey from Johns Hopkins Hospital [51]. The biopsy was from Henrietta Lacks, a 31-year-old AA, who had a very aggressive cervix adenocarcinoma not responsive to radiotherapy. Regrettably, no written informed consent was obtained from Henrietta for the use of her sample for research purposes, an omission that was common practice at the time [51]. In vitro, HeLa cells grew at a rapid doubling time, every 24 h, to which infection by HPV may have been a contributor [52]. When HeLa was established, lab conditions were very different from the ones in current culture cell labs, namely, no laminar-flow cabinets or commercial sterile culture media and sera were available. But the need for large numbers of HeLa cells to grow the polio virus to produce a vaccine led to the setting up of a HeLa production facility at the Tuskegee Institute in Minnesota and of a company to begin producing the medium for HeLa growth 51. The continuing importance of HeLa in research is clear in the number of citations referring to this CCL in PubMed, amounting to 134,217 by 2019 [53].

A few years later, in 1963, another African-derived CCL was established, as the first human continuous hematopoietic CCL (known as RAJI [54]), in Nigeria. The original cells were from a Nigerian patient with Burkitt’s lymphoma infected by Epstein–Barr virus. RAJI was the basis for defining the in vitro culture conditions for hematopoietic CCLs, and it allowed the discovery of new therapeutic strategies targeting leukemic cells [45].

In a paradox to these first representative CCLs, HeLa and RAJI, which were of African origin, current human CCL panels relevant to SSA ethnicities are regrettably very scarce.

#### 4.1.2. Cancer Cell Line Repositories

Given the high demand for HeLa since its establishment, the American Type Culture Collection (ATCC) was set up in 1962 to be a safe repository of CCLs, responsible for the maintenance and provision of CCLs to requesting researchers. However, ATCC ascertained that the 18 supposedly unique human CCLs available at the time were all HeLa cells [55], due to contamination. This was an important setback that led to the establishment of the first rules for CCL contamination and misidentification.

Cross-contamination may happen in the lab due to unnoticed bad practice, such as spread via aerosols, accidental contact, and contaminated reagents, as well as mislabelling and mix-ups in handling [56]. In order to address the cross-contamination problem and decrease its impact on research, norms for the authentication of the studied CCLs were put in place. For human CCLs, short tandem repeat (STR) profiling is the current international reference standard to confirm identity via comparison against the donor tissue and databases containing the STR profiles of commercial CCLs. Another big issue in cell culture is Mycoplasma contamination, reported as early as 1956 [56]. Mycoplasmas possess extremely reduced metabolic capabilities, lack a rigid cell wall, and depend on eukaryotic cells, hence proliferating unrecognized in cell cultures for long periods of time. They are invisible during a routine microscopic inspection of the cells, can pass through conventional microbiological filters, and are not affected by most of the commonly used antibiotics. Although Mycoplasmas do not kill eukaryotic cells, they have considerable effects on them, impacting biological inferences from in vitro analyses. So, every CCL introduced in a lab should be quarantined and tested for mycoplasma contamination, and all active CCLs in the lab should be regularly tested for Mycoplasma contamination [57]. These quality controls are guaranteed by the major CCL repositories: the already mentioned ATCC (USA); the Division of Cancer Treatment and Diagnosis (DCTD) Tumor Repository (USA); the Leibniz-Institute DSMZ; the European Collection of Authenticated Cell Cultures (ECACC); the Japanese Cancer Research Resources Bank (JCRB); the RIKEN BioResource Center (Japan); and the Korean Cell Line Bank (KCLB). These repositories have secured access to a large number of established CCLs, allowing multiple advances in the oncobiology and general biology fields.

#### 4.1.3. Omics Characterization of Cancer Cell Line Panels

With the advent of high-throughput technologies and big data analyses, it has become extremely informative to fully characterize large panels of CCLs for genomics, transcriptomics, proteomics, methylomics, drug response, etc. This extended molecular characterization of large CCL panels is a very powerful tool [58,59]. The first CCL panel to be established was the National Cancer Institute 60 (NCI60) panel, launched in 1988, which included 60 CCLs from nine cancer types (leukemia, melanoma, non-small cell lung, colon, brain, ovary, breast, prostate, and kidney cancers) [36,60]. The main objective of this project was to introduce a new in vitro approach to identify compounds with an effect on tumour cell growth or death [61]. This then revolutionary method tried to overcome the many limitations of drug testing recurring to mice models [61] and introduced automated liquid handling systems to fast-screen thousands of compounds. These 60 CCLs were molecularly characterized for ~120,000 SNP arrays [62], oligonucleotide-base HLA typing [63], mutations in a number of cancer-relevant genes, spectral karyotyping [64], and STR profiling [65]. Since 2000, NCI60 has operated as a service screen and resource, testing new compounds that researchers discover for potential new treatments, currently accomplishing the annual screening of 10,000 compounds [61].

Considerably larger than the NCI60 and other panels established in the meantime (such as JFCR39, with 39 CCLs [66,67]), the Cancer Cell Line Encyclopedia (CCLE; a collaboration between the Broad Institute and the Novartis Institutes for Biomedical Research) project started in 2008, seeking to have around 1000 CCLs from publicly available repositories (now more than 1700 CCLs). This number of CCLs is more robust in capturing the genetic and phenotypic heterogeneity within cancers. The CCLE has made data from its three phases publicly available: phase I obtained Affymetrix SNP 6.0 data, Affymetrix U133 2.0+ expression array data, point mutation profiles with OncoMap 3.0 and hybrid capture exon sequencing of >1600 known or putative cancer genes, and pharmacologic testing across ~500 CCLs for a set of anti-cancer therapeutics [68]; phase II refined the characterization of expressed mRNAs through RNA-seq, genetic alterations through exome sequencing, and quantifying of miRNA, 225 metabolites, bulk Histone H3 tail modifications, and proteins (reverse phase protein array analysis) [48,69,70]; phase III conducted mass spectrometry to quantify the abundance of proteins in 375 CCLs and phosphorylation quantifications [71].

The Wellcome Sanger Institute and the Massachusetts General Hospital-Cancer Center launched the Genomics of Drug Sensitivity in Cancer (GDSC) panel, currently with around 1000 CCLs, many overlapping with the CCLE [72,73]. GDSC comprehensively genetically characterized the incorporated CCLS for whole-exome sequencing; gene expression; copy number alterations; DNA methylation; gene fusions; and microsatellite instability. Iorio et al. [74] demonstrated that GDSC is a rich resource that can link genotypes with cellular phenotypes and then identify therapeutic options. For instance, they found that Temozolamide, used to treat glioblastoma multiforme, also shows activity in *MYC*-amplified colorectal CCLs, a feature that is present in 33% of primary colorectal tumours.

With the introduction of genomic-scale gene-interfering methods, such as RNAi and CRISPR-Cas9, the Wellcome Sanger Institute and Broad Institute have conducted the knockdown/out of thousands of genes in these panels, aiming to directly identify which ones are essential for cancer cell proliferation/survival [75,76]. Many discoveries have already been attained. The development of therapeutics for the deletion of *PTPN2* in tumour cells increased the efficacy of checkpoint blockade by enhancing IFNγ-mediated effects on antigen presentation [77]. Also, Werner syndrome ATP-dependent helicase was identified as a synthetic lethal target in tumours from multiple cancer types with microsatellite instability [76].

Substantial molecular, drug testing, and genetic perturbation data have been made publicly available in publications and user-friendly portals, such as depmap at https://depmap.org/portal/home/#/our-approach (accessed on 17 November 2025), Genomics of Drug Sensitivity in Cancer at https://www.cancerrxgene.org/ (accessed on 17 November 2025), and The Catalogue Of Somatic Mutations In Cancer, COSMIC, at https://cancer.sanger.ac.uk/cosmic (accessed on 17 November 2025). These portals are very useful tools for further analyses, discoveries, and guided selection of the best suitable models to address specific biological questions.

#### 4.1.4. SSA Ancestry Representativeness in CCL Panels

Given the considerable differences in incidences of cancer types and subtypes between ancestries due to germline and somatic heterogeneities, it would be essential to ensure that the in vitro models represent these diversities. Many commercially available CCLs lack information about donor ethnicity. This situation is aggravated in admixed individuals/populations, where ethnic identity does not always match the genetic ancestry. A large study in the USA showed that individuals self-report as AA if it is the majority of their genetic ancestry, while individuals with more than 5% Native American ancestry are most likely to self-identify as Latino [78]. Fortunately, extended genomic characterization of the CCL panels allows us to infer the genetic ancestry of the tumour donors from whom the CCLs were established. Dutil et al. [79] used Principal Component Analysis and Admixture algorithms to estimate the genetic ancestry of 1393 CCLs obtained from COSMIC and CCLE genomic (Affymetrix SNP6.0 Array) data, merging it with 1000 Genomes data for worldwide reference populations [80]. As this array contains nearly one million common SNPs, genotyping the CCLs allows for the screening of germline profiles and ancestry inference of the tumour cell donors from whom the CCLs were established. Dutil et al. [79] confirmed that lots of ambiguities between ethnic identity and genetic ancestry existed in almost half of the CCLs for which ethnicity information was available. The authors found the following distribution of ancestries across CCLs: 62.46% European, 29.18% East Asian, 5.26% AA, 0.86% African, 1.95% Hispanic/Latino, and 0.29% South Asian. So, in terms of the three main human population groups (Figure 2; Appendix A), CCLs can be aggregated as 64.5% EUR, 29.5% Asian, and 6% SSA. Within the SSA ancestry CCL pool, most of these are specific for three cancer types (Figure 2): lung, haematopoietic/lymphoid, and breast. For most cancer types, there are fewer than three CCLs of SSA origin, including for prostate cancer (only one), despite it being so frequent and aggressive in this ancestry. Improving population diversity in CCLs is important to decrease ethnic disparities in research, knowledge, and transferability in the oncobiology field.

#### 4.1.5. Establishment of Cancer Cell Lines: Surpassing Normal Cell Death

The fundamental step to establishing CCLs is surpassing normal cell death. In order to achieve this, cellular senescence must be impeded through a process first characterized by Hayflick and Moorhead [81]. The replicative senescence or mortality stage 1 (M1) of a cell [82] is characterized by the shortening of telomeres and the following stop of cell division. This maximum number of replicative cycles is known as Hayflick’s limit. The cell is unable to proceed with DNA replication but continues to be metabolically active [83]. This replicative senescence can be avoided by, for example, inactivation of the p53 (tumour protein P53) and RB (retinoblastoma) tumour suppressor pathways [83].

Mortality stage 2 (M2) of cultivated cells is the crisis [84]. In this phase, the telomeres are already severely eroded, with chromosomal fusions and translocations [82], leading to genomic instability and a high rate of cell death [85], to add to the already low cell division. During the crisis phase, there is a minor amount of cells (1 in 10^7^ cells) that can mutate to acquire the ability to overcome this high instability state, carry on dividing, and become immortalized [85]. This mortality stage, the crisis, is a second barrier against cancer [86].

Essentially, these two mortality stages consist of a decline in cell division and a high rate of cell death, respectively [84]. The surpassing of these two mortality stages is the basis of all the methods to immortalize cells and has been explored as the breaking point to propagate in vitro primary tumour cells (Figure 3).

##### Conventional Methods of Cell Immortalization

A possible reason for HeLa’s successful in vitro propagation was the original HPV infection. This evidence led to the development of a strategy for fast immortalization, consisting in transducting normal cells with viral DNA oncogenes, the most frequently used being HPV (E6/E7 proteins; [87]) and simian virus 40 (SV40; large T antigen; [88]). Cell lines established in this way have aberrant p53 and RB pathways that normally regulate cell cycle progression. In the case of SV40, it seems that the lifespan of some primary cell types (such as human epithelial cells) is extended, but immortalization is not achieved [89], needing the joint expression of hTERT. Specifically for high-risk HPV, E6 protein induces the human telomerase reverse transcriptase catalytic subunit (hTERT [90]), while E7 protein inactivates the RB pathway, remodels the actin cytoskeleton, and inactivates Rho [91]. E6 and E7 oncogenes have been successful in immortalizing early adult erythroblasts [92] and epithelial cells on the surface of the ovaries [93] but failed in other cases. Also, the use of oncogenic factors transmissible to the host cells raises safety concerns and can induce additional carcinogenesis modifications, leading to extremely aberrant genotypes and phenotypes.

Another conventional strategy for cell immortalization was introduced in 1999, consisting of the ectopic introduction of hTERT, which induces activation of the telomerase. This method was shown to immortalize human fibroblasts [94] and retinal pigment epithelial cells [95]. For other epithelial cells, such as ones derived from human colon biopsies, immortalization could only be achieved when the non-oncogenic protein cyclin-dependent kinase 4 (*CDK4*) was also expressed together with the hTERT catalytic domain [96]. *CDK4* expression bypasses cell culture-associated stresses that frequently lead to premature senescence, while hTERT maintains the length of telomeric DNA and therefore chromosomal stability, helping cells to escape replicative senescence, increasing their proliferation potential [96].

These conventional methods for cell immortalization imply manipulation of the genome, leading to genomic instability, and the consequent irreversible loss of critical biological and genetic characteristics of the original tumour after a few cell passages [97]. Additionally, they present a low success rate, ranging between 1 and 10% [98], and as only a few clones of the original tumour will be immortalized, the established CCLs lack the complex heterogeneity of the original primary tumour [99].

##### Advanced Immortalization Techniques: Conditional Reprogramming

In 2010, Chapman et al. [82] developed a method called conditional reprogramming (CR), which was able to indefinitely extend the life span of primary human keratinocytes in the presence of a Rho kinase (ROCK) inhibitor (Y-27632) in combination with irradiated Swiss 3T3-J2 mouse fibroblast feeder cells (Figure 4). The feeder cells are in growth arrest (usually by irradiation or by the use of the anti-proliferative mitomycin C) and segregate growth factors enabling the growth of the target cells, in addition to detoxifying the medium. The ROCK inhibitor greatly increased the long-term proliferation of primary human keratinocytes, which thus bypassed senescence and became immortal without a detectable cell crisis. The authors showed that the single use of Y-27632 or irradiated fibroblast feeder cells led to senescence in a few passages, for example, 20 to 40 passages for human primary keratinocytes, compared with continuous growth for over 150 passages when both were present [82].

The same research group later successfully applied CR to the more difficult human epithelial cells [100]. They managed to reprogramme primary prostate and mammary cells toward basaloid stem-like phenotypes, which were able to form well-organized spheres in Matrigel. Two days were enough for reprogramming the cell population by the CR method [101], in contrast to the typically longer clonal selection that occurs with conventional methods. The proof that CR directly alters cell growth, rather than selecting for a small subpopulation of stem-like cells, is that even epithelial cells entering the senescence phase were immediately converted to a proliferative state when transferred to the conditions of the CR method [100]. In this way, the phenotypic and genotypic features of the primary tumour are maintained [100], and the developmental potential of the CR cells is also maintained; these cells do not require intricate manipulation to differentiate themselves into the tissue of origin [100,101]. Another advantage is that the induction of CR cells is reversible, as the cells differentiate normally after the removal of Y-27632 and the feeder cells [101].

The mechanism behind the CR method is complex and needs further investigation [99,100,102], but evidence is becoming available. There are parallels with the immortalization induced by HPV [100]: growth with feeder cells is responsible for the induction of hTERT, such as the HPV-16 E6 oncogene, while Y-27632, similarly to the HPV-16 E7 oncogene, leads to alterations in actin/myosin activity and consequently regulates cell proliferation, and it also inactivates Rho13. Both factors were shown to inactivate RB and CDK1 through enhancing their phosphorylation, eliciting a proliferative gene expression profile. Y-27632 reduced Myc-induced apoptosis in primary keratinocytes [103], while CR conditions induced full-length p53 and the natural P53 isoform ∆133P53α to inhibit P53-mediated apoptosis in CR cells [104]. Y-27632 significantly downregulated differentiation-related genes, which were reversible by its removal [82], and it may involve interacting with WNT5A [105] and inhibiting the TGF-β pathway [106]. There is evidence that CR cells maintain stem cell characteristics [107], with Y-27632 upregulating the stem cell markers ∆P63α and CD44 while, in combination with feeder cells, inducing the expression of α6 and β1 integrins and increasing nuclear β-catenin.

Figure 4 resumes the steps of CR application to establish a CCL from a fresh or cryopreserved specimen. By recurring to CR, it is possible to immortalize cells from needle biopsies, cryopreserved tissue, and from fewer than four viable cells [100]. The new CCL can be generated in a relatively short period of time, with an efficiency of up to 90%, far above the 1–10% success rate of traditional immortalization procedures [99]. For instance, a success rate of 83.3% was obtained for bladder cancer [108].

CR can also be used to propagate patient tumour samples for a few divisions, allowing enough material to conduct extensive in vitro characterization. Thus, CR can be a first step before the generation of 2D cultures, spheroids, organoids, and patient-derived xenograft (PDX) models. Hence, CR has been used as a complementary platform for basic, translational, and clinical research. Its success in small biopsy specimens and frozen tissue can potentiate cell-based diagnostics and therapeutics [109]. With CR, the manipulation of epithelial cells ex vivo became easier and more successful.

CR is also attractive for biobanking, as demonstrated by the Human Cancer Models Initiative (HCMI), led by the Cancer Genomics Office of the National Cancer Institute (NCI). HCMI aims to generate 1000 cancer cell models and is committed to addressing “racial disparities”. Again, the intended diversity will rely heavily on AA cohorts, which will continue to be a poor proxy of the highly diverse populations and the complex human–environment interactions across the African continent. By August 2025 (https://hcmi-searchable-catalog.nci.nih.gov/; accessed on 17 November 2025), HCMI had established 22 cell models categorized as “Black or African American” in a total of 321 models (still a bias, with only 7% SSA ancestry). These African models (Appendix A) include nine tissue sources: 10 models are from the colon, 3 from the endometrium, 2 each from the brain and breast, and 1 each from the bronchus/lung, pancreas, rectum, stomach, and uterus. It seems that the gap tends to remain, despite the stated efforts. Given the financial constraints on NCI in the year 2025, and the USA’s political attitude against equity, this initiative may not be successful in changing this bias.

As with all methodologies, particular aspects of CR require special attention. One constraint when using feeder cells is their successful growth arrest, as otherwise potential feeder overgrowth will occur. An effective alternative is the use of conditioned medium derived from irradiated J2 cells, which retains key growth factors secreted by the feeder cells without requiring direct co-culture [99]. Another important consideration is that the ROCK inhibitor can induce cytoskeletal rearrangements, possibly interfering with the migration and invasion properties of tumour cells. This artefact is contradicted by Liu et al. [99], who report that migration and invasion assays could still distinguish normal and tumour cells, indicating that these functions are not blocked. Anyway, a mitigation plan for the culture of CR-derived cell lines without the need for the ROCK inhibitor is the use of Matrigel, at least for a small number of cell passages [100]. A concern with immortalization, including CR, is the potential for genomic and phenotypic drift between the original tumour and the derived cell lines. A careful genomic characterization of six established invasive breast cancer CR cultures (number of passages between 5 and 10) in comparison to the original primary breast tumours by genome-wide array CGH, targeted next-generation sequencing, and global miRNA expression revealed a similar level of copy number alterations (>95% of overlap of cytobands), the retention of specific somatic variants (specifically for the *TP53*, *FLT3*, *JAK3*, *KDR*, *PIK3CA,* and *CDKN2A* genes), and global miRNA profiling clustering of CR and primary tumours [110]. Cultures at a low passage number should minimize genomic and phenotypic divergence.

### 4.2. Advances in Pluripotency and Three-Dimensional Modelling

In the last two decades, in vitro culture has been revolutionized by two promising improvements: the establishment of human-induced pluripotent stem cells (hiPSCs) and advances in three-dimensional (3D) modelling/organoids. Especially when combined, these improvements approximate the in vitro modelling of the in vivo multicellular and spatial organization of tissues and organs. And they are central in the efforts to replace animal models in accordance with the 3R principles (Replacement, Reduction, and Refinement; [111]).

#### 4.2.1. Human-Induced Pluripotent Stem Cells (hiPSCs)

Stem cells (SCs) are undifferentiated or partially differentiated cells that proliferate indefinitely and can differentiate into various types of cells. There are two types of stem cells, embryonic (ESCs) and adult (ASCs). ESCs are derived during early development at the blastocyst stage and are pluripotent, being able to differentiate into any cell type of the three germ layers. ASCs are rare, undifferentiated cells present in many adult tissues, whose role is to maintain and repair the tissue in which they reside. Therefore, ESCs represent a great potential in regenerative medicine [112], but their use raises serious ethical concerns [113,114]. Fortunately, in 2006, somatic cells, named iPSCs, were for the first time successfully induced to become pluripotent by Takahashi and Yamanaka [115]. They were able to reprogramme human adult fibroblasts by introducing four factors, Oct3/4, Sox2, c-Myc, and Klf4, under embryonic stem cell culture conditions. These iPSCs exhibited the morphology, growth properties, and marker genes of ESCs. The authors also verified that when these iPSCs were subcutaneously transplanted into nude mice, they originated tumours containing a variety of tissues from all three germ layers; injected into blastocysts, they contributed to mouse embryonic development. Since this first successful iPSC establishment, which consisted of a transduction technique altering the original genome, other methods have been implemented [102,115], in an attempt to avoid the possibility of genome alteration; examples include techniques based on recombinant cell-penetrating reprogramming proteins [116] and synthetic mRNA expression of embryonic stem genes [117].

iPSCs have been used to differentiate into the three germ layers and then into a large variety of tissues (reviewed in [118]): the endoderm, from which they can be further differentiated in the liver, pancreas, stomach, lung, thyroid and intestine; the mesoderm, to the kidney and heart; and the ectoderm/neuroectoderm, differentiating into the retina, cerebellum, and hippocampus. This pluripotency has been largely explored for studying various musculoskeletal, pulmonary, neurologic, and cardiac phenotypes/diseases (reviewed in [119]). This property is also promising for the use of iPSCs for tissue repair in regenerative medicine, as they are being tested in clinical trials using iPSC-derived dopamine neuron precursors as cell replacement therapy for Parkinson’s Disease [120]. iPSCs definitely have enormous advantages, but as with all technologies, there are limitations that researchers must consider in their experiments. In fact, their cellular reprogramming is incomplete, and the cell type-specific epigenetic pattern from the tissue of origin can persist in reprogrammed iPSCs [121].

The paucity of cross-ethnicity comparisons persists with these sophisticated iPSC models. The iPSCORE resource [122], which aimed to include ethnically diverse individuals, contains 147 Europeans vs. only 4 (and again) AAs, 15 Hispanic individuals, 30 Asians, 6 Indians, 2 Middle Eastern individuals, and 18 mixed-race individuals. HipSci [123] is also extremely biased towards EUR ancestry (64% identified as white, 29% predicted European, 7% without information) and fails to capture global diversity. Some commercial iPSC panels, especially from the USA, may have higher numbers of ethnically diverse iPSCs, but prices to acquire them will be higher than the maintenance costs that are usually required from researchers for access to CCLs/iPSCs from public repositories.

#### 4.2.2. Organoids

Organoids are physiologically relevant 3D models of human organs, better suited to recapitulating human disease pathophysiology than 2D in vitro models and (distant-to-human) animal in vivo models [124]. Organoids can be derived from iPSCs and from isolated organ progenitors (from a patient biopsy, for instance) by employing organ-specific niche factor modulation and an artificially created growth environment where cells can interact in 3D [125]. Thus, organoids mimic communication between cell types and the microenvironment of the organ. These features render organoids an important tool in basic biological research. Current advances in organoid modelling aim at including the immune and vasculature components present in human tissues [125].

The first report of organoids was by Clevers’ group [126]. They used epithelial stem cells of the small intestine to establish crypt–villus organoids using Matrigel. The generation of organoids using adult stem cells is easier, requiring no differentiation, as when using iPSCs, and can be achieved from a patient biopsy [125]. These patient-derived organoids (PDOs) can be extremely useful for pre-drug testing, supporting treatment decisions. In a recent study [127], PDOs from colorectal and gastroesophageal tumours were found to have 100% sensitivity, 93% specificity, 88% positive predictive value, and 100% negative predictive value in forecasting response to targeted agents or chemotherapy in patients. These values support the conclusion that PDOs can be used to simulate cancer behaviour ex vivo and integrate molecular pathology into the decision-making process of early-phase clinical trials. In general organoid-based (non-PDO) testing of drugs, there is increasing evidence that many candidate drugs passing 2D-based screening might fail in 3D, as the cellular microenvironment is a key factor in drug response [128,129].

Another useful application of organoids is biobanking, consisting of collections of genetically and histologically characterized organoid models of disease states with matched controls [130]. In fact, most of the already mentioned HCMI-established cell models categorized as “Black or African American” are organoids (Appendix A). Research groups in the USA are conducting studies with SSA PDOs [131,132,133], and, importantly, a research group from South Africa [134] has implemented a local protocol for the establishment and characterization of South African intestinal PDOs. Unfortunately, other African countries have a wider gap to fill before using and establishing advanced in vitro models locally.

## 5. Potential for Cancer Translation in SSA Patients

It is known that many of the variants with higher differences in frequencies between population groups are located in genes involved in the absorption, distribution, metabolism, and excretion (ADME) of drugs [135,136]. For instance, several genes of the cytochrome P450 (*CYP450*) family, which are involved in phase I drug metabolism, bear variants of potential clinical relevance that display a marked difference in distribution in African compared with Asian and European populations. These variants may be responsible for adverse events following the administration of several internationally commonly used drugs in African individuals. An example is the rs11572103 variant in the *CYP2C8* gene, which reduces the function of the coded enzyme, interfering with the clearance of the drug paclitaxel, widely used for treating several cancers [137]. The derived variant (also known as CYP2C8*2) is common in many African countries, including Mozambique. rs35742686 in the *CYP2D6* gene (CYP2D6*3), which can have a three times higher frequency in San populations compared with Europeans and East Asians, interferes with converting the anti-cancer drug tamoxifen into its anti-estrogenic metabolites [138].

Tests of the interference of these variants with drugs were performed on a pinpoint basis. But the additive or opposing interference of inherited variants in ADME genes in an individual can be now addressed with drug testing in patient-derived advanced models. This opens up a promising new field of ex vivo testing. The first drug sensitivity testing was conducted in South African leukemia patient-derived cells [139], leading to the observation that irinotecan, used in solid tumour treatment, demonstrated efficacy in PBMCs in many patient samples compared to conventional leukemia drugs such as nilotinib.

Thus, genetic ancestry can impact the aggressiveness of disease, type of disease, and response to therapy. Ancestry must begin to be taken into account when evaluating these features through in vitro assays.

## 6. Implementing Oncobiology Studies with Cell Modelling Derived from SSA Patients

The South African research community [140] presented a national framework to direct, sustainably fund, and support translational oncology research with the help of advanced cell culture models. This framework aims at establishing a national community, including the few current research groups in South Africa with advanced cell culture model expertise, in order that these SSA advanced models can be shared with the wider scientific community. This approach enables more locally relevant health questions to be addressed using the most suitable models for the South African context.

As is well known, South Africa is one of the few upper-middle-income countries (UMICs) in SSA, which is reflected in one of the highest investments in science on the continent, further reinforced by external funding from English-speaking funding institutions, such as NIH and The Wellcome Trust [141]. Other SSA countries have much less funding in comparison.

### Case Study: PALOP

The community of Portuguese-speaking African countries (PALOP, to use the Portuguese abbreviation) constitutes an example of lower investment and capacity building. Including Cape Verde (which, in 2024, was classified as a UMIC), Guinea Bissau (a low-income country; LIC), Republic of Equatorial Guinea (UMIC), São Tomé and Príncipe (LMID), Angola (LMID), and Mozambique (LIC), this community sums up over 75 million people, dispersed from off the coast of West Africa to the south of the Atlantic and Indian Ocean coasts. Some years ago, the authors of this review established an informal research network between Angola, Cape Verde, Mozambique, and Portugal. The first research aim was to initiate the acquisition of omics data from cancer patients native to these three PALOP countries, so that this community would not be left out of the incipient omics characterization across the continent. The first studies are beginning to be published [40]. These studies are being conducted by PhD students, both from PALOP countries and Portugal, funded by grants from PALOP and Portuguese funding agencies, who must cooperate, spend some time in both continents, and exchange expertise. These research efforts are coupled with the introduction of simple but important improvements to local diagnosis of cancer in the PALOP, such as the implementation of the point-of-care mRNA STRAT4 breast cancer assay in Cape Verde for sub-classification of breast cancer [142].

The genomic findings we make in PALOP cancer omics will need to be functionally tested so that they can be translated to have a meaningful impact on SSA cancer patient well-being. Thus, the PALOP community should also not miss out on oncobiology modelling, and a plan to circumvent the paucity of SSA-relevant in vitro models is needed. We leveraged this plan against the know-how and infrastructural resources available at i3S-Portugal for the establishment of advanced in vitro models and against obtaining Portuguese-based funding for proof-of-concept research, as well as to reinforce multi-country networking. Visiting the PALOP partners, we evaluated the local technical and infrastructural resources that would allow us to (1) collect fresh specimens from cancer surgeries or biopsies and freeze them under appropriate conditions for transport to Portugal, where they will be used to establish advanced in vitro models, and (2) conduct local cell culture-based experiments with the shared established models. Clinical and scientific personnel were easily engaged in the network. Compliance with local ethical regulations would be straightforward, as bioethics committees were established in these countries in 2000 [143], with local collaborators acting as the main points of contact and local coordinators. PALOP institutions have not yet established strong protocols to ensure data protection for genetic data (in this case, extracted from the advanced models), but the Portuguese partner will share best practices of the European Union General Data Protection Regulation (GDPR; [144]). Significant difficulties were encountered due to limited resources: (1) lack of technical laboratory staff who could take responsibility for promptly collecting fresh samples from the pathology lab and preserving them under appropriate conditions; (2) frequent power outages shutting down the few available −80 °C freezers, which were not supported by reliable backup generators; (3) inadequate biosafety-compliant tissue culture facilities; (4) no possibility of commercially acquiring enough dry ice for inter-country/continental transport of samples; (5) and no reliable transport system to maintain adequate freezing conditions.

Based on these observations, we propose the following roadmap for establishing advanced PALOP cell models, aimed at maximizing timely success:Source of cancer patient tissue: Fresh cancer tissues from PALOP or diaspora patients are to be collected from surgeries and biopsies conducted in hospitals from the Portuguese Health System (SNS, in the Portuguese abbreviation). The PALOP diaspora community residing in Portugal and its descendants have been increasing since 1975, and especially so since the year 2017. Additionally, the SNS has agreements with PALOP health systems for transfer to Portugal and local treatment (including surgeries) of PALOP resident cancer patients. This has the additional advantage of all PALOP countries being represented in the panel, as this diaspora community is multi-ethnic.Establishment of PALOP advanced cell models: Collecting these samples in Portugal allows us the opportunity to establish PALOP advanced cell models in well-equipped Portuguese laboratories while avoiding the difficulties associated with transporting live cells across countries and continents. The CR technique can be used to establish the PALOP CCLs. First efforts should prioritize underrepresented tissues in the few SSA CCLs available in international panels, such as cervix, pancreas, prostate, and kidney tissues. Our very preliminary results indicate that changes to the standard CR protocol [99] may be needed to avoid bacterial contamination (from the sample) in tissues with microbiota (in particular, tissues from the digestive and female reproductive systems).Capacitating PALOP medical doctors, researchers and technicians: Of greatest importance is the training of PALOP medical doctors, researchers, and technicians. There are many protocols established in Portuguese hospitals for the training of PALOP surgeons and oncologists. These visiting PALOP MDs must be involved in the collection of the material for the establishment of PALOP CCLs. Specific training must be established in Portuguese research institutions to train PALOP researchers and technicians in cell culturing techniques and in conducting in vitro experiments. Funding for this training is more easily obtained through PhD grants, which also offer the added benefit of extended (usually four-year) training and the development of an independent researcher.Funding of Portuguese–PALOP partnership: Through a strong network of North–South research partnerships, PALOP scientists can leapfrog technological gaps and build a research ecosystem capable of conducting and leading advanced oncobiology research. This strategy will also allow the PALOP research community to affirm itself within the African continent, on par with English- and French-speaking communities, for instance in the African Organisation for Research and Training in Cancer (AORTIC; https://aortic-africa.org/; accessed on 17 November 2025) and other Health Hubs.

## 7. Conclusions and Future Perspectives

The dramatic increase in cancer incidence in Africa can be effectively mitigated through the timely implementation of straightforward interventions. Fortunately, some countries are already doing so. Broader vaccination against HPV on the continent, and especially if extended to teenage boys, will definitely have a substantial impact. Treatment of schistosomiasis with praziquantel may also have a positive impact in regions where the parasite has a high burden, such as Nigeria and Tanzania, particularly in regions around Lake Victoria, Lake Tanganyika, and the Zambezi River. Greater investment in awareness campaigns about alcohol, tobacco, and ultra-processed foods—including those directed at schools—will improve public health literacy.

However, the genetic disparities between population groups and the incipient uncovered diversity in SSA populations evidence the need for urgent cancer research in the SSA region. The knowledge generated from studying European and Asian cancer patients will not be sufficient when addressing cancer in African populations. The establishment of CCLs from SSA ancestry by a fast and efficient technique such as CR will contribute to decreasing inequalities in cancer research and, hopefully, to improving the discovery of important breakthroughs that will benefit cancer patients in the African continent, and, by extrapolation (by increasing knowledge on oncobiology mechanism heterogeneity), worldwide. The availability of the SSA-CCL panel in a public database will accelerate developments in basic and translational research. Namely, as already demonstrated [74], high-throughput screenings of FDA-approved drug libraries on CCL panels and their correlation with genomic information have provided insightful indications on compound activity across cancer types. Pharmacogenomic approaches to the SSA-CCL panel could help choose drug candidates for further preclinical validation. As many of the drugs included in the libraries will have been used for treatment of other cancers or diseases (drug repurposing), the time and costs of drug testing can be dramatically reduced, providing rapid translation into the clinic. Given the high proportion of infection-related cancer types in Africa, it is expected that drugs interfering with infection will be of importance in the SSA context. Thus, an updated high-throughput screening of drugs in the established SSA-CCL can accelerate an informed transferability of cancer treatments to Africa and its diaspora communities across the globe, improving existing therapies and finding new therapeutic avenues tailored to individuals of African ancestry.

## Figures and Tables

**Figure 1 genes-16-01403-f001:**
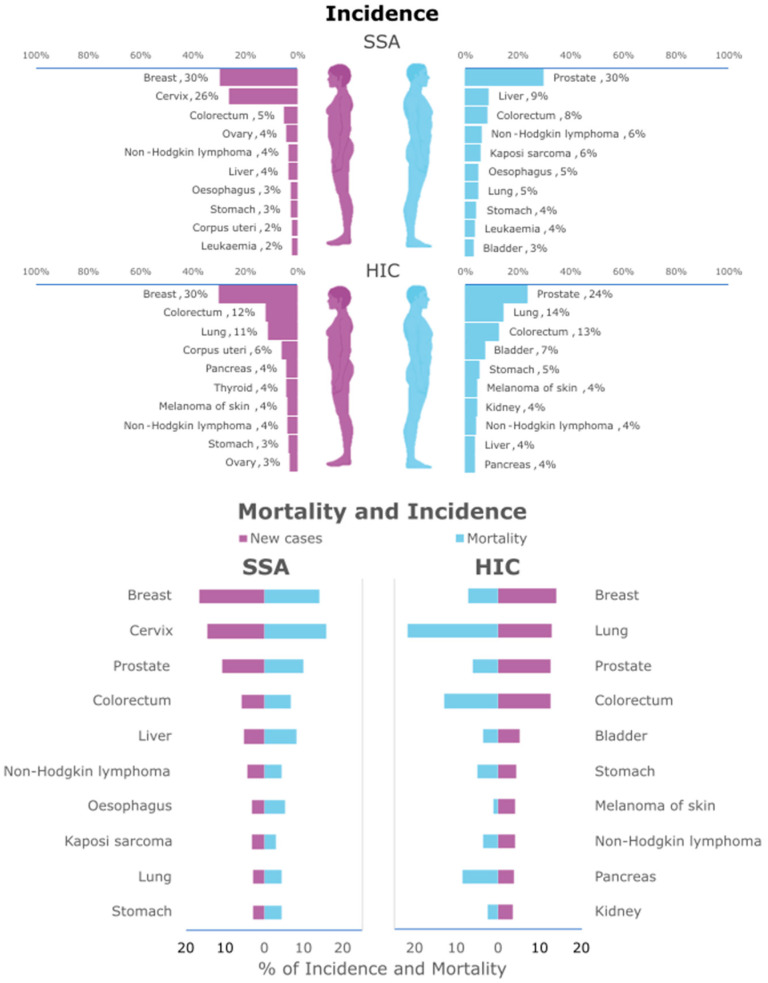
Differences in incidence and mortality rates between SSA and HICs. At the top, differences in cancer incidence between women (on the left) and men (on the right), for the different types of cancer, in SSA and HICs. At the bottom, differences between overall new cases (in purple) and mortality (in blue) in SSA and HICs. Data were obtained from Globocan 2022 [6].

**Figure 2 genes-16-01403-f002:**
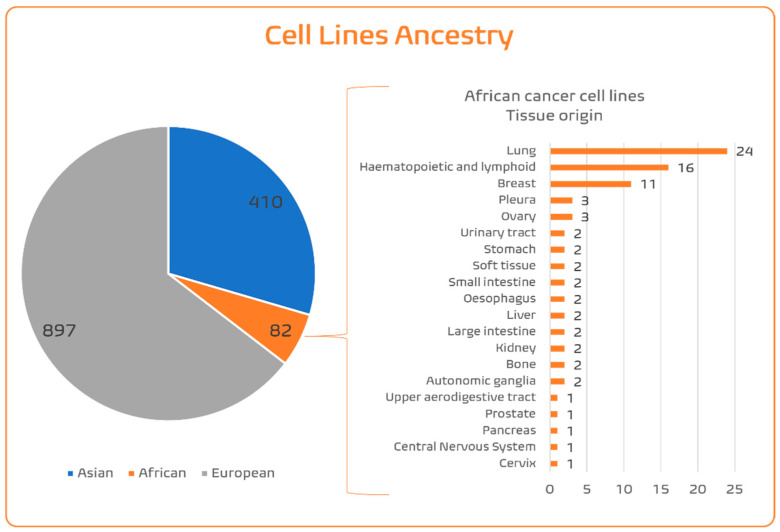
The genetic ancestral origin for the main population groups, Asian, African, and European (includes only individuals with >75% of one ancestral background), of the CCLs included in the CCLE panel [70]. The column graph represents the organ origin of the African CCLs [81].

**Figure 3 genes-16-01403-f003:**
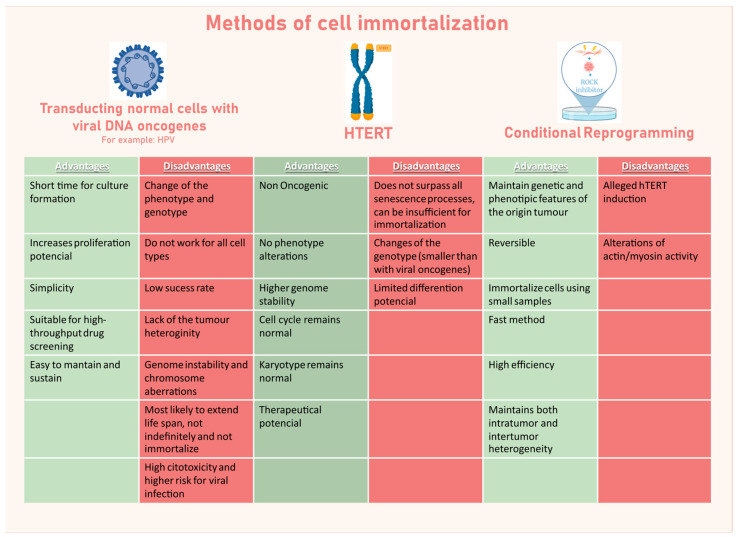
Advantages and disadvantages of diverse methods of cell immortalization, including transduction with viral DNA oncogenes, human telomerase reverse transcriptase (hTERT), and conditional reprogramming.

**Figure 4 genes-16-01403-f004:**
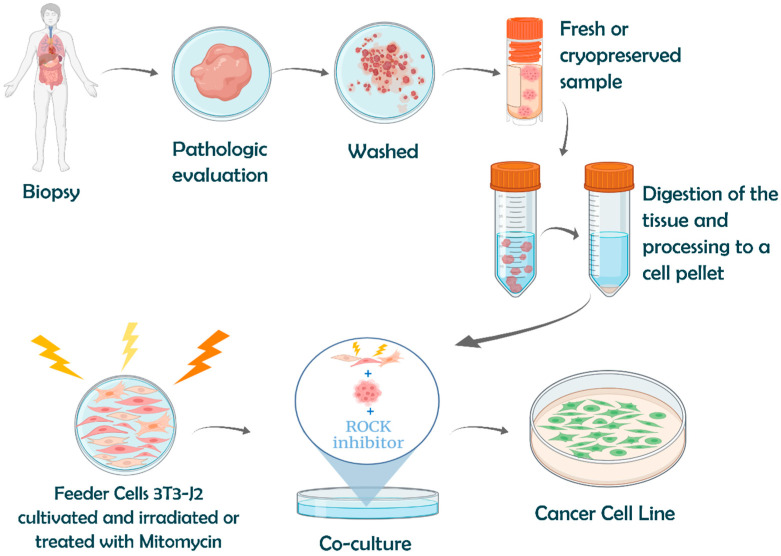
Graphical representation of the conditional reprogramming method. A sample from the patient biopsy is evaluated by a pathologist to separate the tumour from normal tissue. After washing, the tissue can be cultured fresh or cryopreserved. Before culture, the tissue is digested. The cell pellet obtained is co-cultured with previously irradiated feeder cells. Created with Biorender.com.

## Data Availability

No new data were created or analyzed in this study.

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
