# Peer review of "Addressing Ancestral Underrepresentation in Oncobiology: The Need for Sub-Saharan African-Specific In Vitro Models"

_genes, 2025, doi:10.3390/genes16121403_

Round 1

Reviewer 1 Report

Comments and Suggestions for Authors

The authors present a review arguing for development of Sub-Saharan Africa (SSA)–specific in vitro cancer models (CCLs, CR, iPSC/organoids) and describe practical barriers and a PALOP case study. The manuscript is useful but requires clarifications, additional discussion of ethics/regulation, methods limitations and some corrections.

-The Abstract states “cancer-incidence is predicted to increase 70% per year”. Is this phrase correct? Please provide exact numbers, dates and references.

-Expand on key technical limitations / caveats: dependency on mouse 3T3-J2 feeder cells (xenogenic components), potential feeder overgrowth, alteration of cytoskeleton via ROCK inhibitor affecting invasion/migration assays, and whether CR alters key molecular readouts long-term. Cite experimental studies that quantify genomic drift vs primary tumor. The manuscript mentions some of these but does not emphasize them; explicit discussion and suggested mitigation strategies (e.g., feeder-free CR variants, mycoplasma/feeder testing, short-term assays) are required.

-For claims of “up to 90% efficiency” please provide sources and specify tissue types and sample conditions (fresh vs cryopreserved). If this number derives from HCMI or specific CR publications, cite them.

-Discuss more local regulatory restrictions.

-The manuscript calls for action but is light on measurable next steps. Add a short table or numbered roadmap (e.g., 1. create SSA CCL inventory; 2. prioritize cancer types & regions; 3. standardize consent/MTA; 4. capacity building; 5. data deposition) with target milestones, responsible actors, and minimal resource estimates. This will greatly increase the manuscript’s practical impact.

-The review claims ~6% SSA representation in public CCLs and lists tissues with sparse coverage. Please include a concise supplementary table listing known SSA-origin CCLs/organoids (name, tissue, origin country, repository, genomic data availability), with references (e.g., Dutil et al. and CCLE/COSMIC entries). This supports reproducibility of the claim.

-Statements implying that increasing SSA-CCL panels will directly result in near-term clinical benefit (e.g., immediate design of early-phase trials) should be softened or supported with concrete prior examples of successful translational cascades. Add one or two documented examples where CCL expansion directly led to clinical trials or drug repurposing in low-resource settings.

-A few sentences are imprecise (e.g., “SSA will testify a major increase” — please rephrase). Run a brief language pass for clarity and grammar.

-Consider noting limitations of inferring ancestry from tumor DNA (somatic alterations can confuse inference).

-Consider moving the PALOP case study into a separate short section titled “Case study: PALOP” to increase readability.

Comments on the Quality of English Language

moderate editing

Author Response

The authors present a review arguing for development of Sub-Saharan Africa (SSA)–specific in vitro cancer models (CCLs, CR, iPSC/organoids) and describe practical barriers and a PALOP case study. The manuscript is useful but requires clarifications, additional discussion of ethics/regulation, methods limitations and some corrections.

1- The Abstract states “cancer-incidence is predicted to increase 70% per year”. Is this phrase correct? Please provide exact numbers, dates and references.

We thank the reviewer for these helpful comments. The “per year” was a typographical error, and should “is predicted to increase 70% between 2012 and 2030”. We took this opportunity to recheck values across the text, and decided to change this sentence to “is predicted to increase 140% between 2022 and 2050” (without reference in the Abstract), and changed sentences in the beginning of Introduction for “It is predicted [6] that, if no intervention is done at the moment, SSA will experience a major increase in cancer mortality from 763,843 annual deaths in 2022 to almost 1.8 million by 2050.” with [6] being the reference to Globocan 2022 data.

2- Expand on key technical limitations / caveats: dependency on mouse 3T3-J2 feeder cells (xenogenic components), potential feeder overgrowth, alteration of cytoskeleton via ROCK inhibitor affecting invasion/migration assays, and whether CR alters key molecular readouts long-term. Cite experimental studies that quantify genomic drift vs primary tumor. The manuscript mentions some of these but does not emphasize them; explicit discussion and suggested mitigation strategies (e.g., feeder-free CR variants, mycoplasma/feeder testing, short-term assays) are required.

We appreciate the reviewer comment and agree this topic deserves consideration. So, we have expanded the short previous paragraph (and changed it to the end of the section) to:

“As with all methodologies, particular aspects of the CR require special attention. One constraint when using feeder cells is their successful growth arrest, as otherwise potential feeder overgrowth will occur. An effective alternative is the use of conditioned medium derived from irradiated J2 cells, which retains key growth factors secreted by the feeder cells, without requiring direct co-culture [103]. Another important consideration is the ROCK inhibitor can induce cytoskeletal rearrangements, possibly interfering with the migration and invasion properties of the tumour cells [103]. This artefact is contradicted by Liu et al. [103] report that migration and invasion assays could still distinguish normal and tumor cells, indicating that these functions were not blocked. Anyway, a mitigation plan for the culture of CR derived cell lines without the need for ROCK inhibitor is the use of Matrigel, at least for a small number of cell passages [104]. A concern with immortalization, including CR, is the potential for genomic and phenotypic drift between the original tumor and the derived cell lines. A careful genomic characterization of six established invasive breast cancer CR cultures (number of pas-sages between 5 and 10) in comparison to the original primary breast tumours, by genome-wide array-CGH, targeted next generation sequencing and global miRNA ex-pression, revealed a similar level of copy number alterations (>95% of overlap of cyto-bands), the retention of specific somatic variants (specifically for the TP53, FLT3, JAK3, KDR, PIK3CA and CDKN2A genes), and global miRNA profiling clustering of CR and primary tumours [114]. Cultures at low passage number should minimize genomic and phenotypic divergence. “

3- For claims of “up to 90% efficiency” please provide sources and specify tissue types and sample conditions (fresh vs cryopreserved). If this number derives from HCMI or specific CR publications, cite them.

We added an example of this high success rate. See page 14, new sentence “For instance, a success rate of 83,3% was obtained for bladder cancer [117].”

4- Discuss more local regulatory restrictions.

Thanks for calling our attention to this topic. We included the following information: “Compliance with local ethical regulations would be straightforward, as bioethics committees were established in these countries in 2000 [145], with local collaborators acting as the main points of contact and local coordinators. PALOP institutions have not yet established strong protocols to ensure data protection for genetic data (in this case, extracted from the advanced models), but the Portuguese partner will share best practices of the European Union General Data Protection Regulation (GDPR; [146]).”

5- The manuscript calls for action but is light on measurable next steps. Add a short table or numbered roadmap (e.g., 1. create SSA CCL inventory; 2. prioritize cancer types & regions; 3. standardize consent/MTA; 4. capacity building; 5. data deposition) with target milestones, responsible actors, and minimal resource estimates. This will greatly increase the manuscript’s practical impact.

Thank you so much for this comment. Now the section of the manuscript was reorganized as a roadmap and reads like this: “Based on these observations, we propose the following roadmap for establishing advanced PALOP cell models, aimed at maximizing timely success:

1- Source of cancer patient tissue: the fresh cancer tissues from PALOP or its diaspora patients to be collected from surgeries and biopsies conducted in hospitals from the Portuguese Health System (SNS, in the Portuguese abbreviation). The PALOP diaspora community residing in Portugal, and its descendants, have been increasing since 1975, and especially so since the year 2017. Additionally, SNS has agreements with PALOP health systems for the transfer to Portugal and local treatment (including surgeries) of PALOP resident cancer patients. This has the additional advantage of all PALOP countries being represented in the panel, as this diaspora community is multi-ethnic.

2- Establishment of the PALOP advanced cell models: Collecting these samples in Portugal allows the opportunity to establish the PALOP advanced cell models in well-equipped Portuguese laboratories, while avoiding the difficulties associated with transporting live cells across countries and continents. The CR technic can be used to establish the PALOP CCLs. First efforts should prioritize underrepresented tissues in the few SSA CCLs in international panels, such as cervix, pancreas, prostate and kidney. Our very preliminary results indicate that changes to the standard CR protocol [101] may be needed to avoid bacterial (from the sample) contamination in tissues with micro-biome (in particular, the digestive and female reproductive systems).

3- Capacitating PALOP medical doctors, researchers and technicians: Of most importance is the training of PALOP medical doctors, researchers and technicians. There are many protocols established in Portuguese hospitals for the training of PALOP surgeons and oncologists. These visiting PALOP MDs must be involved in the collection of the material for the establishment of PALOP CCLs. Specific training must be established in Portuguese research institutions to train PALOP researchers and technicians in cell culturing techniques and in conducting in vitro experiments. Funding for this training is more easily obtained through a PhD grant, which also offers the added benefit of extended (usually four-year) training and the development of an independent researcher.

4- Funding of the Portuguese-PALOP partnership: Through a strong network of North-South research partnerships, PALOP scientists can leapfrog technological gaps and build a research ecosystem capable of conducting and leading advanced oncobiology research. This strategy will also allow the PALOP research community to affirm itself within the African continent, in par with English- and French-speaking communities, for instance in the African Organisation for Research and Training in Cancer (AORTIC; https://aortic-africa.org/) and other Health Hubs.”

6- The review claims ~6% SSA representation in public CCLs and lists tissues with sparse coverage. Please include a concise supplementary table listing known SSA-origin CCLs/organoids (name, tissue, origin country, repository, genomic data availability), with references (e.g., Dutil et al. and CCLE/COSMIC entries). This supports reproducibility of the claim.

Following suggestions of reviewer 1 and 3, we have now included two Supplementary Tables, intending to concisely summarize information for SSA-origin CCLs/organoids. We agree with the reviewers that these Supplementary Tables will be useful for the readers of the manuscript.

7- Statements implying that increasing SSA-CCL panels will directly result in near-term clinical benefit (e.g., immediate design of early-phase trials) should be softened or supported with concrete prior examples of successful translational cascades. Add one or two documented examples where CCL expansion directly led to clinical trials or drug repurposing in low-resource settings.

The reviewer is right that we should tone down these statements. This part of Conclusions reads now as “The availability of the SSA-CCL panel in a public database will accelerate developments in basic and translational research. Namely, as already demonstrated [77], high-throughput screenings of FDA-approved drug libraries on CCL panels and their correlation with genomic information have provided insightful indications on compound activity across cancer types. Pharmacogenomic approaches in the SSA-CCL panel could help choose drug candidates for further preclinical validation. As many of the drugs included in the libraries will have been used for treatment of other cancers or diseases (drug repurposing), the time and costs of drug testing can be dramatically reduced, and provide rapid translation into the clinic. Given the high proportion of infection-related cancer types in Africa, it is expected that drugs interfering with infection will be of importance in the SSA context. Thus, an updated high-throughput screening of drugs in the established SSA-CCL can accelerate an informed transferability of cancer treatments to Africa and its diaspora communities across the globe, improving existing therapies and finding new therapeutic avenues tailored to the African ancestry.”

8- A few sentences are imprecise (e.g., “SSA will testify a major increase” — please rephrase). Run a brief language pass for clarity and grammar.

We changed this sentence to “SSA will experience a major increase in cancer mortality […]”, and we conducted a global language check-up.

9- Consider noting limitations of inferring ancestry from tumor DNA (somatic alterations can confuse inference).

This is not a concern in the ancestry inference conducted by Dutil et al., that we used in the review. Dutil et al. used Affymetrix SNP6.0 Array data genotyped in the CCLs. This array contains 906,600 SNPs that were used in the ADMIXTURE algorithm, merged with 1000 Genomes data. These SNPs were selected from HapMap genomic profiles from worldwide populations, having a high average minor allele frequency: 19.6% in HapMap Caucasians; 18.2% in HapMap Asians; and 20.6% in HapMap Africans. Even if genotyped in the CCLs, they will vastly represent the germline profiles of the donors of the tumor cells from which CCLs were established. We added the following information on the manuscript: “Dutil et al. [84] used Principal Component Analyses and Admixture algorithms to estimate the genetic ancestry from 1,393 CCLs departing from COSMIC and CCLE genomic (Affymetrix SNP6.0 Array) data, and merging with the 1000 Genomes data for worldwide reference populations [85]. As this array contains nearly one million common SNPs, genotyping the CCLs allows for the screening of the germline profiles and the ancestry inference of the tumour cell donors from whom the CCLs were established.”

10- Consider moving the PALOP case study into a separate short section titled “Case study: PALOP” to increase readability.

Thank you for calling our attention to this. We now split headings as “4. Implementing oncobiology studies with cell modelling derived from SSA patients” where we introduce initiatives in South Africa, and “4.1. Case study: PALOP” specific for PALOP.

11- Comments on the Quality of English Language: moderate editing

We did revise the English throughout the manuscript.

Reviewer 2 Report

Comments and Suggestions for Authors

Review of the article "Addressing ancestral underrepresentation in oncobiology: the need for Sub-Saharan African-specific in vitro models"

In my opinion, the article is interesting and valuable, but the Authors should try to point out the reason for the dramatic increase in cancer incidence in the SSA populations in order to point out in what way to try to fight cancer in SSA.

Based on the latest advances in cancer research, "cancer transformation (i.e., a change in normal cell-fate to cancerous/atavistic cell fate) occurs as a result of loss of control over functionalities of the unicellular layer, resulting in a loss of control over atavistic functionalities" (https://doi.org/10.3390/ijms23074017) and "cancer transformation can occur as a result of huge disturbances in functionalities of the multicellular layer that normally control activity of atavistic functionalities".
In my opinion, the Authors should try to add (for example, before the Conclusions and future perspectives section) the interpretations of possible reasons for such a high increase in the number of cancer incidences in Sub-Saharan Africa (SSA). From the article, it is clearly visible that, among others, (a) specific for SSA viruses (as it is in lines 88-93: "28.7% of SSA cancers are due to infectious agents, especially viruses as human papilloma-viruses (HPV), herpesvirus, hepatitis viruses (B and C), Epstein–Barr virus and human immunodeficiency virus, but also Helicobacter pylori and Schistosoma haematobium. This infection-related proportion is twice the value of 12.5% cancers associated with traditional risk factors such as smoking, alcohol and unhealthy diet [12], although this last fraction is probably on the rise in Africa with the changes in lifestyle."), (b) specific climate in SSA and (c) changes in lifestyle (smoking, alcohol, and unhealthy diet) can affect multicellular layer functionalities, causing loss of control over functionalities of the unicellular layer, leading to cancer transformation and then cancer progression.
I suggest that the Authors expand this interpretation and present it in the article, which should increase its value and potentially indicate how to fight cancer in the SSN populations. This interpretation can also help with establishing in vitro models to test hypotheses (as it is in lines 23-25, i.e., "There is an urgent need to improve the genomic characterization of SSA tumour samples, and also to establish suitable in vitro models to test hypotheses").

Author Response

We thank the reviewer for the comments. As the reviewer acknowledges, we already indicate the reasons for the increase of cancer incidence in Africa. To extend on this topic would deserve a dedicated Review manuscript. This is not the scope of our manuscript. Nevertheless, we have included a paragraph on Conclusions and Future Remarks that addresses interventions that could be implemented in Africa, currently or in the near future, that would have a positive impact to oppose the increasing cancer incidence in Africa. Now this section begins with: “The dramatic increase in cancer incidence in Africa can be effectively mitigated through the timely implementation of straightforward interventions. Fortunately, some countries are already doing so. Definitely, a broader vaccination against HPV in the continent, and especially if enlarged to teenage boys, will have a substantial impact. Treatment of schistosomiasis with praziquantel may also have a positive impact in regions where the parasite has a high burden, such as Nigeria and Tanzania, particularly in regions around Lake Victoria, Lake Tanganyika, and the Zambezi River. Greater in-vestment in awareness campaigns about alcohol, tobacco, and ultra-processed foods—including those directed at schools—will improve public health literacy.”

Reviewer 3 Report

Comments and Suggestions for Authors
  1. he paper is generally well-structured, with logical progression from epidemiology to genomics and in vitro modeling. However, some sections (particularly 3.1 and 3.2) could benefit from concise subheadings to enhance readability for multidisciplinary audiences.
  2. The manuscript repeatedly emphasizes that only ~6% of publicly available cancer cell lines (CCLs) represent African ancestry, yet lacks a table or figure consolidating this data by cancer type or repository. The authors should include quantitative summaries (e.g., per database such as CCLE, ATCC, HCMI) to substantiate their claim.
  3. The most recent key references (e.g., GLOBOCAN 2022, Lancet Oncology 2022) are now outdated given ongoing cancer genomics efforts across Africa. The authors should integrate 2023-2024 updates from WHO’s African Cancer Registry and recent studies in Nature Medicine and NPJ Precision Oncology.
  4. While CR is well detailed, the discussion should be broadened to compare other next-generation immortalization and culture systems (e.g., organ-on-chip, co-culture with immune components, CRISPR-based reprogramming). A short comparative table summarizing advantages and limitations would improve scholarly balance.
  5. The review discusses SSA genomic heterogeneity but does not clearly connect it to phenotypic or therapeutic differences in vitro. Adding examples of ancestry-specific drug metabolism genes (e.g., CYP450 variants) or response patterns would better justify the need for SSA-derived models.
  6. The historical overview of cell lines (e.g., HeLa, RAJI) is highly informative but overly detailed for a review aimed at oncobiology translation. Condensing this section and shifting emphasis toward actionable research gaps will make the manuscript more focused and impactful.
  7. The PALOP section presents a strong conceptual framework. To enhance the section’s practicality and completeness, please consider including specific supporting data, such as sample sizes, tissue types, or preliminary success rates of conditional reprogramming from PALOP sources, to better illustrate the feasibility of implementation.

Author Response

The paper is generally well-structured, with logical progression from epidemiology to genomics and in vitro modeling.

1- However, some sections (particularly 3.1 and 3.2) could benefit from concise subheadings to enhance readability for multidisciplinary audiences.

Thank you for this comment. We added new subheading “3.1.3. Omics characterization of cancer cell lines panels”; “3.2. Advances in pluripotency and three-dimensional modelling”; “3.2.1 Human induced pluripotent stem cells (hiPSCs)”; “3.2.2 Organoids”.

2- The manuscript repeatedly emphasizes that only ~6% of publicly available cancer cell lines (CCLs) represent African ancestry, yet lacks a table or figure consolidating this data by cancer type or repository. The authors should include quantitative summaries (e.g., per database such as CCLE, ATCC, HCMI) to substantiate their claim.

Following suggestions of reviewer 1 and 3, we have now included two Supplementary Tables, intending to concisely summarize information for SSA-origin CCLs/organoids. We agree with the reviewers that these Supplementary Tables will be useful for the readers of the manuscript.

3- The most recent key references (e.g., GLOBOCAN 2022, Lancet Oncology 2022) are now outdated given ongoing cancer genomics efforts across Africa. The authors should integrate 2023-2024 updates from WHO’s African Cancer Registry and recent studies in Nature Medicine and NPJ Precision Oncology.

We acknowledge the reviewer’s comment. We decided to recur to GLOBOCAN 2022 because we wanted to compare cancer incidence and mortality between Africa and high-income countries – so this global concise database offers comparable data. We now checked throughout that we only used values from GLOBOCAN 2022. From Lancet Oncology 2022, we just picked up summary sentences. The most recent African Cancer Registry Network concise report (Cancer in Sub-Saharan Africa III) was published in 2019, reporting data from 2010 to 2017. Permission to access to their current online database must be analysed by a committee, and would not be conceded timely for this revision. Anyway, we agree that efforts by this group must be acknowledge, and included a reference “Hopefully, as other SSA countries begin improving their vaccination programmes [5], a significant impact will be observable in the next years SSA cancer registry conducted by the African Cancer Registry Network (AFCRN - https://afcrn.org/index.php).”

4- While CR is well detailed, the discussion should be broadened to compare other next-generation immortalization and culture systems (e.g., organ-on-chip, co-culture with immune components, CRISPR-based reprogramming). A short comparative table summarizing advantages and limitations would improve scholarly balance.

We decided to include iPSC and organoids because, although limited, there are some examples of SSA iPSC and organoids. To include other advanced culture technics, for which we could not find any study with SSA samples, would be a substantial deviation from the scope of this review. It became even more difficult to do this extension, as reviewer 1 asked us to include more considerations on regulations and roadmap to begin establishing CCLs and in vitro studies in Africa, and the extension we decided worth doing in response to this reviewer next question. We hope the reviewer agrees with our decision.

5- The review discusses SSA genomic heterogeneity but does not clearly connect it to phenotypic or therapeutic differences in vitro. Adding examples of ancestry-specific drug metabolism genes (e.g., CYP450 variants) or response patterns would better justify the need for SSA-derived models.

Old section 3.1.3, new section 3.1.4. “SSA ancestry representativeness in CCL panels” made this connection. But we agree with the reviewer that this topic is important and decided to include it in a new independent section “4. Potential for cancer translation in SSA patients”. This new section reads as: “It is known that many of the variants with higher differences in frequencies between population groups are located in genes involved in the absorption, distribution, metabolism, and excretion (ADME) of drugs [137, 138]. For instance, several genes of the cytochrome P450 (CYP450) family, which are involved in phase I drug metabolism, bear variants of potential clinical relevance that display a marked difference in distribution in African, compared with Asian and European populations. These variants may be responsible for African adverse events for several internationally commonly used drugs. An example is the rs11572103 variant in CYP2C8 gene, which reduces the function of the coded enzyme, interfering with clearance of the drug paclitaxel, widely used for treating several cancers [139]. The derived variant (also known as CYP2C8*2) is common in many African countries, including Mozambique. The rs35742686 in CYP2D6 gene (CYP2D6*3), that can have a three times higher frequency in San populations compared with Europeans and East Asians, interferes with converting the anti-cancer drug tamoxifen into its anti-estrogenic metabolites [140].

Tests of the interference of these variants with drugs were done at a pinpoint basis. But the additive or opposing interference of the inherited variants for the ADME genes in an individual can be now addressed with drug testing in patient-derived advanced models. This opens up a promising new field of ex-vivo testing. A first drug-sensitivity testing was conducted in South African leukemia patient-derived cells [141], leading to the observation that irinotecan, used in solid tumour treatment, demonstrated efficacy in PBMCs in many patient samples compared to conventional leukemia drugs such as nilotinib.

Thus, genetic ancestry can impact the aggressiveness of disease, type of disease, and response to therapy. Ancestry must begin to be taken into account when evaluating these features through in vitro assays.”

6- The historical overview of cell lines (e.g., HeLa, RAJI) is highly informative but overly detailed for a review aimed at oncobiology translation. Condensing this section and shifting emphasis toward actionable research gaps will make the manuscript more focused and impactful.

We condensed this section as suggested by the reviewer.

7- The PALOP section presents a strong conceptual framework. To enhance the section’s practicality and completeness, please consider including specific supporting data, such as sample sizes, tissue types, or preliminary success rates of conditional reprogramming from PALOP sources, to better illustrate the feasibility of implementation.

Also following recommendations of reviewer 1, we substantially reformatted the new independent section “5.1. Case study: PALOP”, and include some practical insights in the roadmap. Please have a look into this section.

Round 2

Reviewer 1 Report

Comments and Suggestions for Authors

The manuscript was significantly improved. I appreciate the correction of the “per year” typographical error and the effort to ensure consistency across the manuscript.

However, I would like to ask for further clarification regarding the statement that cancer incidence “is predicted to increase 140% between 2022 and 2050.” Based on GLOBOCAN 2022 projections, a ~140% increase appears to correspond to mortality projections in sub-Saharan Africa (from 763,843 deaths in 2022 to almost 1.8 million by 2050), not incidence. Global and regional incidence increases predicted by GLOBOCAN 2022 are substantially lower. To ensure accuracy, please confirm whether the 140% figure refers to incidence or mortality, and adjust the wording accordingly. If incidence data are intended, kindly verify the exact projected values from GLOBOCAN 2022 and update the percentage to match those projections.

Author Response

We checked and it is incidence. Please have a look into: the data in this website: https://gco.iarc.fr/tomorrow/en/dataviz/bars?populations=903&years=2050&types=0

Thank you again for all the helpful comments.

Reviewer 3 Report

Comments and Suggestions for Authors

No more comments

Author Response

Many thanks for all the helpful inputs to our paper.